# The Effect of Ketamine on Endoplasmic Reticulum Stress in Rats with Neuropathic Pain [note 1]

**DOI:** 10.3390/ijms24065336

**Published:** 2023-03-10

**Authors:** Eun-Hye Seo, Liyun Piao, Eun-Hwa Cho, Seung-Wan Hong, Seong-Hyop Kim

**Affiliations:** 1Korea mRNA Vaccine Initiative, School of Medicine, Gachon University, Incheon 21565, Republic of Korea; 2Department of Infection and Immunology, School of Medicine, Konkuk University, Seoul 05030, Republic of Korea; 3Department of Anesthesiology and Pain Medicine, Konkuk University Medical Center, School of Medicine, Konkuk University, Neudong-ro (Hwayang-dong), Seoul 05030, Republic of Korea; 4Department of Medicine, Institute of Biomedical Science and Technology, School of Medicine, Konkuk University, Seoul 05030, Republic of Korea

**Keywords:** ER stress, neuropathic pain, ketamine, N-methyl-D-aspartate receptor subtype 2B, activating transcription factor-6

## Abstract

This study aimed to investigate the effects of ketamine, an N-methyl-D-aspartate (NMDA) receptor antagonist, on endoplasmic reticulum (ER) stress in rats with neuropathic pain (NP). NP was induced in rats through ligation and transection of the sciatic nerve. After confirmation of NP, the animals were randomly divided into ketamine and control groups. The ketamine group was administered 50 mg/kg of ketamine at 15, 18, and 21 days after surgery. The expression of NMDA receptor subtype 2B (NR2B) and ER stress markers in the spinal cord (L5) was evaluated. The ipsilateral side of the surgery in the ketamine group was less sensitive to mechanical and cold stimulations. The expression of NR2B on the ipsilateral side was significantly lower in the ketamine group than in the control group (18.93 ± 1.40% vs. 31.08 ± 0.74%, *p* < 0.05). All markers for ER stress on the ipsilateral side of the surgery in both groups had higher expression than those on the contralateral side. The expression of activating transcription factor-6 (ATF-6) on the ipsilateral side was significantly lower in the ketamine group than in the control group (*p* < 0.05). Systemic administration of ketamine inhibited the expression of NMDA receptors and improved NP symptoms. Among the markers of ER stress, the therapeutic effect of ketamine is associated with the inhibition of ATF-6 expression.

## 1. Introduction

Neuropathic pain (NP) is defined as pain resulting from damage to or dysfunction of the nervous system that is often characterised by allodynia and hyperalgesia. Despite robust efforts, the mechanism of action of NP remains unclear. After nerve damage, the N-methyl-D-aspartate (NMDA) receptors are activated, triggering pro-apoptosis [1]. Therefore, nonselective NMDA receptor antagonists, such as ketamine, have been used to treat various types of NP and have shown effectiveness [2,3,4].

The endoplasmic reticulum (ER) is an organelle in eukaryotes involved in cell homeostasis, including the transport of folding proteins. Only properly folded proteins can be transported from the ER to the Golgi apparatus. Under harmful conditions, an increase in unfolded proteins leads to ER stress, leading to cell apoptosis. Recently, some studies have shown an association between NP and ER stress [5,6,7]. However, the association between NMDA receptors and ER stress has not been explored.

We hypothesised that ketamine, an NMDA receptor antagonist, might influence the expression of ER stress in NP. Therefore, this study investigated the effect of ketamine on ER stress in rats with NP.

## 2. Results

Eighteen rats were enrolled in the study and evenly allocated to the two groups. Surgery for NP in all experiments was successfully performed without any complications. Ketamine and normal saline were administered without any adverse events.

Before surgery, the contralateral and ipsilateral sides of the surgery in both groups showed no differences in withdrawal response to mechanical and cold allodynia (Figure 1). However, after surgery, the ipsilateral side in both groups showed significantly lower thresholds for mechanical stimulation and a significantly earlier withdrawal response to cold stimulation than the contralateral side (Figure 1). After treatment with ketamine or normal saline 15, 18, and 21 days after surgery, the withdrawal response to von Frey filaments and dry ice on the contralateral side of the surgery showed no significant changes in the two groups (Figure 1). However, on the ipsilateral side, mechanical and cold stimulation responses were lower in the ketamine group than those in the control group (mechanical, control vs. ketamine: 1.07 ± 0.36 g vs. 6.98 ± 1.22 g at D15_After_, *p* < 0.05; 1.34 ± 1.49 g vs. 7.13 ± 1.80 g at D18_After_, *p* < 0.05; 1.52 ± 1.09 g vs. 7.37 ± 1.57 g at D21_After_, *p* < 0.05; cold, control vs. ketamine: 8.33 ± 1.86 s vs. 23.67 ± 1.97 s at D15_After_, *p* < 0.05; 10.50 ± 2.51 s vs. 23.33 ± 4.80 s at D18_After_, *p* < 0.05; 10.33 ± 4.80 s vs. 23.83 ± 1.60 s at D21_After_, *p* < 0.05; Figure 1 and Figure 2).

The expression of NMDA subtype 2B(NR2B) on the contralateral side was similar between the two groups (10.82 ± 2.35% in the control group vs. 12.44 ± 1.07% in the ketamine group, *p* = 0.59). The expression of NR2B on the ipsilateral side in both groups was higher than that on the contralateral side. The control group showed a significant difference, but the ketamine group did not (31.08 ± 0.74% in the control group, *p* < 0.05; 18.93 ± 1.40% in the ketamine group, *p* = 0.07). The expression of NR2B on the ipsilateral side was significantly lower in the ketamine group than that in the control group (*p* < 0.05) (Figure 3).

All markers for ER stress on the ipsilateral side in both groups had higher expression than those on the contralateral side. Only Activating transcription factor-6 (ATF-6) on the ipsilateral side in the control group was significant (*p* < 0.05). The expression of ATF-6 on the ipsilateral side was significantly lower in the ketamine group than that in the control group (*p* < 0.05) (Figure 4).

## 3. Discussion

In the present study, the systemic administration of ketamine inhibited the expression of NMDA receptors and ameliorated NP symptoms. It also inhibited the expression of ATF-6, but not Protein kinase R-like endoplasmic reticulum (PERK) or CCAAT-enhancer-binding protein homologous proteins (CHOP), in response to ER stress.

ER stress has been suggested to be involved in nervous system disorders such as Alzheimer’s [8,9] and Parkinson’s diseases [10], among others [11,12]. Recently, several studies have demonstrated the involvement of ER stress in NP. For example, Inceoglu et al. demonstrated that ER stress in the peripheral nervous system is a significant driver of NP. Furthermore, they revealed that the activation of the signal transduction cascade for ER stress generated site-restricted pain, which was reversed by an ER stress blocker. In addition, they showed that ER stress levels and NP-related behaviour in diabetic rats with NP were reduced by ER stress blockers [13]. Moreover, Ge et al. demonstrated that intrathecal thapsigargin, an ER stress inducer, reduced the mechanical thresholds for pain in healthy rats with induced ER stress; however, intrathecal tauroursodeoxycholic acid, an ER stress inhibitor, significantly reduced ER stress in the dorsal horn of the spinal nerve with the alleviation of nociceptive behaviour in rats with peripheral nerve injury [6]. Additionally, Zhang et al. demonstrated that ER stress might be involved in inducing and maintaining NP in rats with L5 spinal nerve ligation-induced NP [5]. Furthermore, Khangura et al. reported ER stress as a new target for NP [14].

Although previous studies have demonstrated the therapeutic effect of ketamine on NP, the mechanism of ketamine has been focused on only NMDA receptors. Therefore, the present study, focusing on the association between ketamine, an NMDA antagonist, and ER stress in NP, was distinct from the previous studies. In this study, we first revealed that the amelioration of NP-related allodynia using NMDA receptor antagonists such as ketamine was involved in inhibiting ER stress, although the exact pathway was not evaluated. The involvement of NMDA receptors and ER stress in other nervous system disorders has been previously investigated. For instance, Dong et al. reported that interleukin-1β, implicated in the aetiology of Alzheimer’s disease, disturbs intracellular calcium homeostasis via an NMDA receptor-mediated mechanism, triggering neuronal apoptosis by enhancing ER stress [15]. In addition, Costa et al. reported that ER stress via NMDA receptor-dependent mechanisms is involved in hippocampal dysfunction [16].

Remarkably, in the present study, the expression of ATF-6 only among ER stress markers was inhibited by ketamine. The expression of other ER stress markers was also increased in the spinal cords of other NP models [5,13]. However, Zhang et al. showed that the release of ATF-6 from the spinal cord was observed in rats with NP induced by L5 spinal nerve ligation. Furthermore, they demonstrated that intrathecal injection of small interfering ribonucleic acids for ATF-6 to inhibit the production of ATF-6 alleviated mechanical hypersensitivity in rats with L5 spinal nerve ligation-induced NP [5]. Additionally, Zhou et al. found a higher expression of ATF-6 in a formalin-induced pain model in rats. The pain was relieved with a decrease in ATF-6 expression after treatment with the ER stress inhibitor 4-phenylbutyric acid [17]. Therefore, the degree of ATF-6 expression, a marker of ER stress, is associated with the degree of NP.

In this study, ketamine (50 mg/kg, a relatively high dose) was administered intraperitoneally. Investigations with lower doses of ketamine (up to 30 mg/kg) showed no improvement in NP [18,19]. Therefore, we selected a dose of 50 mg/kg. One consideration remains: Wang et al. did not show any improvement in NP 24 h after administering 50 mg/kg ketamine [18], but Zhou et al. showed an improvement in NP 3 h after administering the same dose [17]. In the present study, we demonstrated an improvement in allodynia 1 h after ketamine injection. Therefore, the duration of the ketamine effect and injection frequency should be considered in determining the treatment effect. In the present study, no differences in the expression of NR2B and ER stress markers between the contralateral and ipsilateral sides after surgery were observed in the ketamine group. This may be due to the systemic effects of ketamine.

Among the various NMDA receptors, NR2B in the spinal cord plays an important role in NP development and therapeutic targeting [20,21,22]. Therefore, we selected NR2B for this study. Although NR2B expression was mainly observed in neuronal cells, the evaluation of ER stress and NMDA receptors in specific cells would help clarify the role of ER stress in the expression of NMDA receptors.

## 4. Materials and Methods

### 4.1. Animals

This study was conducted after obtaining approval from the Konkuk University Institutional Animal Care and Use Committee (KU18097). Six to eight-week-old male Sprague Dawley rats were purchased from Orient Bio (Seongnam, Republic of Korea). All experiments were performed in accordance with the National Institutes of Health (NIH) guidelines for the care and use of laboratory animals. All the animals were housed in two groups in cages with free access to water and food. The cages were maintained under a standard 12 h light and dark cycle (lights on at 7:00 and lights off at 19:00) and a temperature of 25 °C. All animals were acclimated to the experimental conditions for 7 days before NP surgery, and all experiments were performed during the daytime. Before NP surgery, all animals were assessed for NP and allodynia. Surgery for NP was performed after the assessment confirmed the absence of NP. At 3, 7, and 14 days after NP surgery, NP expression was evaluated using the same techniques as those before surgery. If sufficient NP expression was not observed in the assessment, the animal was excluded from the experiment. After confirmation of NP expression, the animals were randomly divided into ketamine and control groups. At 15, 18, and 21 days after NP surgery, ketamine or normal saline was intraperitoneally administered according to the designated groups. Ketamine (50 mg/kg) mixed with normal saline (0.5 mL) was injected into rats in the ketamine group, while an equal volume of normal saline was injected into those in the control group. The assessment of NP was performed before and 60 min after administering ketamine or normal saline.

### 4.2. Assessment of NP

NP was assessed by mechanical and cold allodynia using von Frey filaments and dry ice, respectively. The von Frey filaments comprised forces of 0.6, 1.0, 1.4, 2.0, 4.0, 6.0, 8.0, 11.0, and 15.0 g. The animals were placed in a transparent test cage with a wire mesh metal floor for 30 min before the test. The rigid tip of the von Frey filaments was perpendicularly applied to the skin of the lateral plantar area of the left hind paw until it bent, starting with a force of 2 g. If a response such as paw withdrawal or licking occurred, a weaker force was applied using the Dixon up–down method. If a response did not occur, a stronger force was applied using the Dixon up–down method. A maximum force of 15.0 g was assigned as the threshold. Cold allodynia was measured using dry ice at −80 °C. The animals were placed in a transparent cage with a glass floor for 30 min before testing for environmental adaptation. Dry ice was ground into a fine powder using a hammer. Thereafter, the dry ice powder was filled into a 10-mL syringe and compressed until it could not be compressed further. Using a syringe plunger, a pellet of dry ice was pushed 20–30 mm past the tip of the syringe and gently and firmly applied to the glass underneath the hind paws of the rats, after which the withdrawal response was measured.

### 4.3. Surgery for NP

Anaesthesia was induced using a 5% volume of isoflurane (JW Pharmaceutical, Seoul, Republic of Korea) in conjunction with oxygen (300 mL/min) and nitrous oxide (700 mL/min). After induction of anaesthesia, the animals were moved to the surgical platform, and anaesthesia was assessed by pinching the hindfoot. Next, the tongue was pulled out using forceps, with the animals in a supine position. After intubation with a 1.77-inch-long 16 gauge 4.5 cm catheter (BD Biosciences, San Jose, CA, USA), the correct intubation position was confirmed by checking symmetric chest expansion. A ventilator (Harvard Apparatus, Holliston, MA, USA) was connected to the intubation catheter and set to (1) a fraction of inspired O_2_ (FiO_2_) of 0.5, (2) an inspiratory flow rate of 170 mL/min, (3) tidal volume of 6 mL/kg, and (4) respiratory rate of 80 breaths/min. Anaesthesia was maintained with isoflurane, using a 4.0% vaporiser, via an intubation catheter. After intubation and ventilator setting, the position was gently changed to the prone position. The depth of the intubation catheter was rechecked in the prone position. After confirming the proper position for intubation, surgery was performed. The left leg of each animal was fastened to the platform with tape, and the hair around the left thigh was shaved. After sterilisation with 70% alcohol around the shaved thigh, the sciatic nerve, with three branches of the common peroneal, tibial, and sural nerves, was exposed through a 2 cm incision at the posterior side along the heel. The two nerves, except the sural nerve, were ligated with silk 5-0 and transected 2 mm distal to the ligation. After transection, the muscle and skin were sutured, and the incision site was disinfected.

### 4.4. NMDA Receptor Expression

After completing all tests 21 days after NP surgery, the animals were anaesthetised by intraperitoneal injection with 5 μg/g xylazine (Rompun^®^; Bayer Korea Ltd., Seoul, Republic of Korea) and 50 μg/g Zoletil 50 (a combination of tiletamine and zolazepam; Virbac Laboratories, Carros, France). Subsequently, paraformaldehyde (4% in 0.1 M phosphate-buffered saline (PBS)) was infused via catheterisation of the left ventricle, and the blood was drained into the incision at the right atrium. A spinal cord segment (L5) was isolated and fixed in a 4% formalin solution overnight at 4 °C. Following deparaffinisation and hydration of the tissue, the L5 segment was sliced using a microtome. Next, the sliced sections were placed on slides and blocked with 3% hydrogen peroxide for 10 min at room temperature, after which they were incubated with primary antibodies, including anti- NR2B (ab216621; 1:250; Abcam, Cambridge, UK) for 2 h at 37 °C. After incubation, the slides were washed with 0.1 M PBS thrice for 2 min and subsequently incubated with a goat anti-rabbit secondary antibody (PV-9001; 1:500; ZSGB-BIO; OriGene Technologies Inc., Rockville, MD, USA) for 30 min at 37 °C. Afterwards, the sections were stained with diaminobenzidine (DAB kit; ZSGB-BIO; OriGene Technologies, Inc.) and counterstained with haematoxylin for 30 s at room temperature. Thereafter, images were captured under a light microscope (Olympus Corporation, Tokyo, Japan). Quantitative image analysis of the relative optical density was performed using ImageJ with Java 8 (NIH, Bethesda, MD, USA).

### 4.5. Immunohistochemistry Assay for ER Stress Markers

Immunochemical staining for ATF-6, PERK, and CHOP as markers of ER stress in L5 was performed. Briefly, tissues from the organs were embedded in paraffin, and transverse paraffin sections (5 μm thick) were mounted on saline-coated slides. For immunohistochemistry, the mounted sections were washed in 0.01 M PBS containing 1% Tween 20 (pH 7.4, PBS-T) and then immersed in 2% normal horse serum in PBS for 2 h at 37 °C. They were then incubated overnight at 4 °C with ATF-6 (1:3000; Novus Biologicals, Englewood, CO, USA), PERK (1:200; Bioss Antibodies, Woburn, MA, USA), and CHOP antibodies (1:150; Santa Cruz Biotechnology, Dallas, TX, USA) in PBS containing 1% bovine serum albumin. After incubation, the sections were washed thrice in PBS for 5 min, after which they were incubated in biotinylated horse–anti-mouse immunoglobulin G (1:200; Boster Bio, Pleasanton, CA, USA) in PBS for 2 h at room temperature. Thereafter, they were washed in PBS thrice for 5 min and incubated in avidin–biotin–peroxidase complex solution (ABC, 1:100, Boster Bio) for 2 h at room temperature, after which they were rinsed again thrice in PBS for 5 min. After the staining procedure, the sections were counterstained with haematoxylin and then dehydrated using ethanol and xylene before coverslipping with Permount^®^. Visualisation was performed by incubating the tissue for 2 min in 0.04% 3,3-diaminobenzidine (DAB; Sigma-Aldrich, St. Louis, MO, USA) containing 0.01% H_2_O_2_. Rat immunoglobulin G (1:200; Biomeda Corporation, Foster City, CA, USA) was used instead of the primary antibody as a negative control. To quantify immunostaining, the numbers of ATF-6-, PERK-, or CHOP-positive cells in the same area were counted using ImageJ with Java 8.

### 4.6. Statistics

The primary outcome was the expression of ER stress markers, including ATF-6, PERK, and CHOP, on the ipsilateral side of the surgery. In a pilot study with three rats in the control group, the expressions of ER stress markers were 81.74 ± 1.30%, 8.09 ± 0.36%, and 7.24 ± 0.98% for ATF-6, PERK, and CHOP, respectively. A 20% reduction in the expression of ER stress markers was considered significant. The calculated sample sizes for the primary outcomes were two for ARF-6, three for PERK, and nine for CHOP in each group from the pilot study, with an α of 0.05 and a power of 0.8.

Intra- and inter-group differences were analysed using an unpaired *t*-test and two-way repeated ANOVA using GraphPad Prism software (ver. 5.01; GraphPad Software, La Jolla, CA, USA). A *p*-value < 0.05 was considered statistically significant. Data are presented as mean ± standard deviation.

## 5. Conclusion

In conclusion, the systemic administration of ketamine inhibited the expression of NMDA receptors and ameliorated NP symptoms. The therapeutic effect of ketamine was associated with inhibiting the expression of ATF-6, but not that of PERK or CHOP, during ER stress.

## Figures and Tables

**Figure 1 ijms-24-05336-f001:**
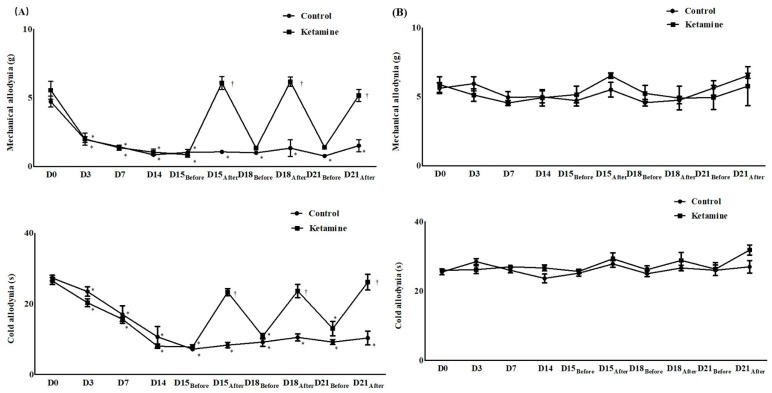
Assessment of allodynia for neuropathic pain before and after the treatment of ketamine or normal saline. (**A**) Ipsilateral side for the surgery and (**B**) contralateral side for the surgery. Abbreviations: control, control group; ketamine, ketamine group; D0, before the surgery; D, day after the surgery; before, before the treatment of ketamine or normal saline; after, 60 min after the treatment of ketamine or normal saline. * *p* < 0.05 compared with D0 in intra-group variance. ^†^
*p* < 0.05 compared with the control group.

**Figure 2 ijms-24-05336-f002:**
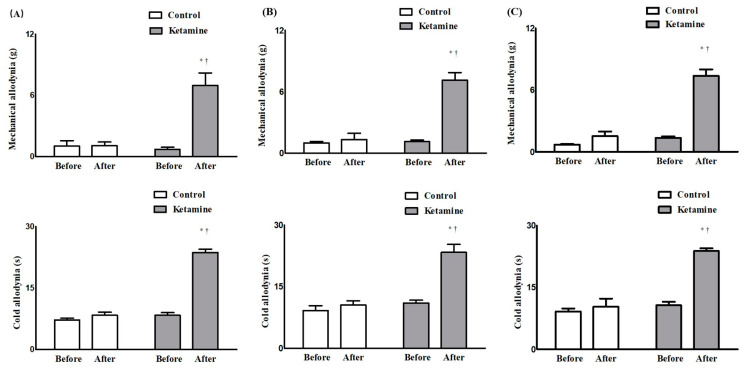
Assessment of allodynia for neuropathic pain at the ipsilateral side for the surgery before and after the treatment of ketamine or normal saline on 15 days (**A**), 18 days (**B**), and 21 days (**C**) after the surgery. * *p* < 0.05 compared with before treatment. ^†^
*p* < 0.05 compared with the control group.

**Figure 3 ijms-24-05336-f003:**
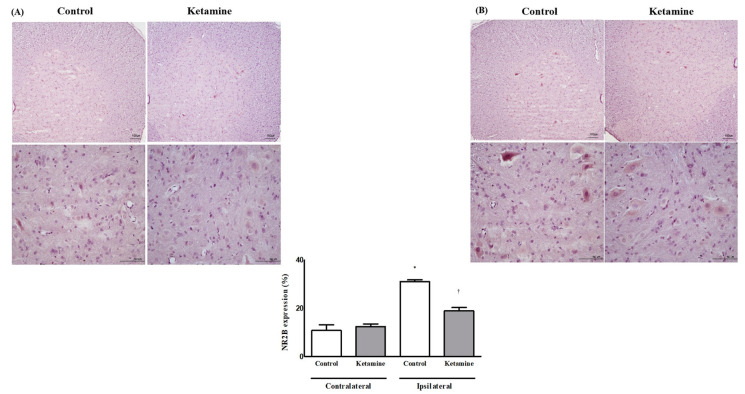
The expression of N-methyl-D-aspartate receptor subtype 2B (NR2B) in the spinal cord. (**A**) Contralateral side for the surgery with a magnified figure, ×40 (below) and (**B**) ipsilateral side for the surgery with a magnified figure, ×40 (below). Abbreviations: control, control group; ketamine, ketamine group. * *p* < 0.05 compared with the contralateral side in each group. ^†^
*p* < 0.05 compared with the control group.

**Figure 4 ijms-24-05336-f004:**
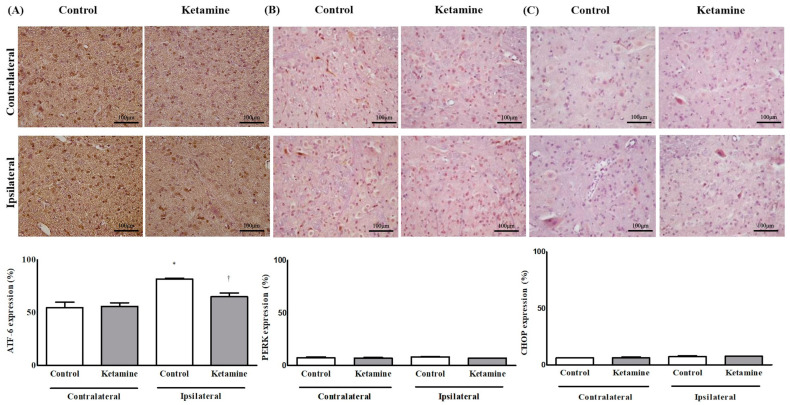
The immunohistochemistry assay for the expression of endoplasmic reticulum stress in the spinal cord. (**A**) Activating transcription factor-6 (ATF-6), (**B**) Protein kinase R-like endoplasmic reticulum (PERK), and (**C**) CCAAT-enhancer-binding protein homologous proteins (CHOP). Abbreviations: contralateral, contralateral side for surgery; ipsilateral, ipsilateral side for the surgery; control, control group; ketamine, ketamine group. * *p* < 0.05 compared with the contralateral side in each group. ^†^ *p* < 0.05 compared with the control group.

## Data Availability

The datasets are available from the corresponding author upon reasonable request.

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
