# Peer review of "The Effect of Ketamine on Endoplasmic Reticulum Stress in Rats with Neuropathic Painâ€"

_ijms, 2023, doi:10.3390/ijms24065336_

Round 1
Reviewer 1 Report
The reason I accepted this paper is as follows.
1. Authors evaluated the effect of ketamine on ER stress in rat with neuropathic pain. They found that therapeutic effect of ketamine was associated with ATF-6 expression in rat with neuropathic pain. Although there are many studies for neuropathic pain, the association between neuropathic pain and ER stress has not been revealed clearly. Especially, authors revealed the effect of ketamine on ER stress. Previous study about the association between neuropathic pain and ER stress have focused on the mechanism of neuropathic pain. However, the present study focused on the therapeutic effect. Therefore, the present study had the scientific value.
2. The sample size determination in the present study was conducted from the pilot study. Therefore, the findings had the statistical significance.
3. Therefore, I thought that the present study was suitable to be published.
4. I was not able to decide whether the manuscript was suitable in English form because I am not native English.
Author Response
At first, I thank the editors and referees of the “International Journal of Molecular Sciences” for taking their times to review of my paper, entitled “The effect of ketamine on endoplasmic reticulum stress in rat with neuropathic pain”.
I have made some corrections and clarifications in the manuscript after going over the referee’s comments. The changes are summarized below and the corrected or newly added sentences were expressed with red-color in the manuscript.
Academic Editor
- After reading the paper and glancing through the figures, it is my impression that ER stress is not involved in this process. The evidence are weak and real biochemical analysis was performed. I am therefore not supporting a publication in this issue.
: We are sorry for academic editor’s opinion that the manuscript was not enough to explain the effect of ketamine on endoplasmic reticulum (ER) stress in rat with neuropathic pain. The present study showed that ketamine inhibited the expression of N-methyl-D-aspartate (NMDA) receptors and improved neuropathic pain as demonstrated in previous studies. However, there has been no study to evaluate the association between ketamine and ER stress in neuropathic pain. To confirm the association between ketamine and ER stress in neuropathic pain, all makers of ER stress such as activating transcription factor-6 (ATF-6), protein kinase R-like endoplasmic reticulum (PERK), and CCAAT-enhancer-binding protein homologous protein (CHOP) were checked in the present study. Only ATF-6 expression was significantly inhibited by ketamine. Therefore, we concluded that the therapeutic effect of ketamine on neuropathic pain was associated with inhibiting the expression of ATF-6, but not that of PERK or CHOP, during ER stress. In the present study, we first revealed that the amelioration of neuropathic pain-related allodynia using NMDA receptor antagonists such as ketamine was involved in inhibition of ER stress, although the exact pathway was not evaluated. Therefore, we thought that the present study gave new information to the readers of International Journal of Molecular Sciences.
We added the several sentences to get the results concretely in Discussion
Reviewer-1
- The reason I accepted this paper is as follows. 1. Authors evaluated the effect of ketamine on ER stress in rat with neuropathic pain. They found that therapeutic effect of ketamine was associated with ATF-6 expression in rat with neuropathic pain. Although there are many studies for neuropathic pain, the association between neuropathic pain and ER stress has not been revealed clearly. Especially, authors revealed the effect of ketamine on ER stress. Previous study about the association between neuropathic pain and ER stress have focused on the mechanism of neuropathic pain. However, the present study focused on the therapeutic effect. Therefore, the present study had the scientific value. 2. The sample size determination in the present study was conducted from the pilot study. Therefore, the findings had the statistical significance. 3. Therefore, I thought that the present study was suitable to be published.
: I thanked for your review.
- I was not able to decide whether the manuscript was suitable in English form because I am not native English.
: Before the submission, the manuscript was reviewed by English-editing service.
We added the certificate for English-editing service.
Reviewer-2
- However, the microscope magnification is not visible in Figure 3.
: We changed Figure 3.
- They stated that the ipsilateral side of the surgery in the ketamine group was less sensitive to mechanical stimulation. However, in the results section, the ipsilateral side in both groups showed significantly lower thresholds for mechanical stimulation and a significantly earlier withdrawal response to cold stimulation than the contralateral side. Edit your abstract.
: We changed the sentence as below.
→ The ipsilateral side of the surgery in the ketamine group was less sensitive to mechanical and cold stimulations.
I hope the revised manuscript will better meet the requirements of the “International Journal of Molecular Sciences” for publication.
I thank you again for the constructive review by the referees.
Sincerely yours,
Seong-Hyop Kim, M.D., Ph.D.

Reviewer 2 Report
In this manuscript, it's a very interesting and trending paper that provides useful information by endoplasmic reticulum stress, the effect of ketamine is associated with the inhibition of activating transcription factor-6 expression.
However, the microscope magnification is not visible in Figure 3.
Please confirm the additional review comment. The review comment is written below. Review comment: There have been many efforts to reveal the mechanism for occurrence of neuropathic pain. Although the association between ER stress and neuropathic pain have been reported in previous studies, the present study showed not only the mechanism of neuropathic pain through ER stress but also therapeutic mechanism of ketamine on neuropathic pain through ER stress. Ketamine has been clinically used as one of popular therapies for neuropathic pain. Therefore, the present study provided the basis of the clinical use of ketamine for neuropathic pain, although authors did not show any association between NMDA receptor and ER stress in neuropathic pain. Abstract They stated that the ipsilateral side of the surgery in the ketamine group was less sensitive to mechanical stimulation. However, in the results section, he ipsilateral side in both groups showed significantly lower thresholds for mechanical stimulation and a significantly earlier withdrawal response to cold stimulation than the contralateral side. Edit your abstract.
Author Response

(The authors gave the same response as above.)
